# Evaluating the Usefulness of the Modified Swimming Anaerobic Sprint Test (SAST) Based on the Relationship with the 100- and 200-m Freestyle

**Łukasz Wądrzyk [1],\*** , **Robert Staszkiewicz [2]** and **Marek Strzała [1]**

[1] Department of Water Sports, Faculty of Physical Education and Sport, University of Physical Education, al. Jana Pawla II 78, 31-571 Krakow, Poland; marek.strzala@awf.krakow.pl

[2] Department of Biomechanics, Institute of Biomedical Sciences, University of Physical Education, 31-571 Krakow, Poland; robert.staszkiewicz@awf.krakow.pl

\* Correspondence: wadrzyk504@gmail.com; Tel.: +48-12-683-11-05

**Abstract:** The construction of specialized physiological tests for swimmers is often based on those previously used in versions on land. So far, no attempt has been made to adapt the RAST test to conditions of an aquatic environment. The aim of the study was to evaluate the relationship between the results from the 6 × 25 m front crawl test (Swimming Anaerobic Sprint Test—SAST) and 100- and 200-m freestyle performance. In a group of 22 swimmers (9 girls, 13 boys, mean age 17.2 years, body mass 67.6 kg, body height 1.76 m), the SAST was performed (10-s interval between repetition). To determine speed (v), stroke rate (SR), stroke length (SL), and stroke index (SI), the test and races were recorded with the use of cameras. The analysis included times from the competition and fatigue indices (the decrease in SI during the SAST). Analysis of variance was performed, and a correlation matrix was created considering variables from the SAST and competition. The indices describing the 200-m event differed from those recorded for the SAST and 100 m freestyle race. The SAST results (v, SL, SI) were strongly correlated with the 100- and 200-m performances. Fatigue indices have not been shown to correlate with competition performance. The SAST may be useful in the context of predicting results for the 100- and, to a limited extent, 200-m freestyle races.

**Keywords:** swimming; biomechanics; video analysis; movement

## 1. Introduction

In swimming competitions taking place at pools, we may distinguish sprint (50–100 m), middle- (200–400 m), and long-distance races (800–1500 m) [1]. Each of the competitions places different requirements on the athletes, mainly in terms of physiology [2,3]. As a result, success in swimming events is often determined by various factors, e.g., the effectiveness of using various energy systems [4]. In the case of short distances, good results are achieved mainly by athletes capable of coping with the efforts on an anaerobic lactic acid basis, with a high ability to tolerate this type of fatigue [5]. In addition, large body size (often associated with greater height and muscle mass) and the potential to generate high mechanical power are also important in the swimming sprint [6]. In long-distance events, the result mainly depends on efficiency of the aerobic energy supply system and the economy of swimming (e.g., low energy cost) [2]. In middle-distance races, swimming performance is determined by a combination of determinants for shorter and longer events [7]. It should be added that, apart from the aforementioned physiological factors, the level of mastery of the swimming technique has significant impact on sports result, which depends, among others, on body structure [8,9].

Bearing these deliberations in mind, it seems reasonable to assume a diagnosis regarding the state of the swimmer's physical fitness with parallel technical control of task performance, i.e., by determining the values of selected kinematic indices. The quality

of performed movements can be assessed by applying basic kinematic indices, such as stroke length (SL) and stroke rate (SR) [10]. Some authors also distinguish stroke index (SI = v × SL), which is considered to be an exceptionally good indicator of swimming efficiency [11,12].

The diagnosis of physiological disposition level allows for identifying factors limiting sports performance of swimmers [8]. In training practice, these diagnostics can be carried out, inter alia, by test swimming. Monitoring outcome components should, therefore, be an integral part of a planned training program. The construction of specialized water tests for swimmers is often based on those previously used in versions on land. An example of adapting land tests is tethered swimming, being an equivalent of the Wingate test [13] or the progressive swim test, analogous to the progressive run test [14].

One of the tests used to assess anaerobic capacity on land is the Running Anaerobic Sprint Test (RAST), in which the athletes cover six 35-m runs in the shortest time possible, with a 10-s interval between each repetition [15]. The maximal and mean mechanical power, as well as the fatigue index [16], are determined based on the time required to complete successive runs and the athlete's body mass. The design of this test (short efforts of maximum intensity) meets the design criteria for anaerobic capacity tests. Low costs make this test of practical use even in youth or amateur sports [16]. In this regard, the RAST test has been shown to be very useful in diagnosing indicators of anaerobic capacity, as evidenced by a high correlation between the results of this test and those obtained for the Wingate test [17]. Therefore, it seems reasonable to modify this test to the conditions of an aquatic environment and determine the relationship between its results and that for swimming, especially in the case of short- and middle-distances.

The facts described above justify the research objective: the need to assess the usefulness of the modified RAST test in the form of a 6 × 25 m (hereinafter referred to as the Swimming Anaerobic Sprint Test—SAST) in relation to swimming results, especially in teenage swimmers, among whom less technologically advanced methods are implemented. The main aim of the research was to determine the relationship between the SAST results and those obtained for the short- (100-m) and middle-distance freestyle (200-m). The analysis covered changes in speed, swimming stroke length, stroke rate, and stroke index. The hypothesis was deemed verified based on the results of the aforementioned relationship: the SAST test is a useful tool for predicting sports results over a short distance, i.e., the 100-m freestyle. It was also assumed that its use to predict the result for middle-distances (200-m freestyle) is limited. The last direction of the analysis was to assess whether the basic kinematic variables describing a swimmer's movement in further repetitions of the SAST assume similar values as their identical indices from the final parts of the 100- and 200-m freestyle races.

## 2. Materials and Methods

### 2.1. Participants and Anthropometric Measurements

A total of 22 volunteers aged 16 to 19 (mean age 17.2 ± 0.7 years, body mass 67.6 ± 8.4 kg, body height 1.76 ± 0.08 m) participated in the study. The participants comprised 9 girls and 13 boys. The subjects represented the 2nd sports class, appropriate for their age, in accordance with the standards of the Polish Swimming Federation. The body height of the subjects was determined using the Radwag WPT 100/200 OW stadiometer. Body mass was measured using the Tanita BC1000 body composition analyser device. Anthropometric measurements were made when the subjects were in swimming suits. The main measurements were carried out under training conditions in a 25-m swimming pool (stage 1) and a few days later at regional control competitions (stage 2) in a 50-m swimming pool.

### 2.2. Stage 1

The essential element of this part of measurements was the SAST (Swimming Anaerobic Sprint Test). Before its start, swimmers performed a standard 10-min warm-up on land, followed by the aforementioned anthropometric measurements. Immediately after that,

the second part of the warm-up took place in water. It was planned and supervised by a coach working with the athletes on a daily basis. After its completion and a short period devoted to resting, the subjects began the SAST.

The test consisted of covering, in the shortest possible time, a distance of 25 m six times while performing the front crawl stroke. There was a 10-s passive rest between repetitions. Each measurement segment began from the water, after a push-off on the turning wall without submerging and using underwater undulatory swimming. The starting signal was a whistle sound given by the person controlling the interval time. The course of the test was recorded using three GoPro Hero Black 7 devices (GoPro Inc., San Mateo, CA, USA) in 'recording mode' at a frequency of 60 frames/s. They were placed on stable tripods at a distance of approx. 1.5 m from the side wall of the pool. One camera (in 'Superview' mode) was set in the center at a height of about 3 m in a way that allowed the recording of a competitor's five complete cycles. The other two cameras (in "Linear" mode) were placed at a height of 1 m, each 7.5 m from the turn walls. This setting resulted in the distance between the two devices being 10 m. Synchronization of the cameras was done with the use of the Swim Start Synchro (Opti.Eng, Krakow, Poland) device, emitting a simultaneous light signal visible in the lens of both cameras. This action made it possible to calculate the time to cover each 10-m section on the basis of the moment when the head passed the 7.5 m line from each turn wall. In order to facilitate identification of this section, permanent markings visible in the lens of the cameras were placed on both sides of the pool.

### 2.3. Stage 2

The next stage of the research was analysis of the performance at swimming competitions. The subjects participated in two regional competitions, competing at 100- and 200-m distances in the freestyle. These competitions were recorded using one GoPro Hero Black 7 camera with a frequency of 60 frames/s in the 'Linear' recording mode. The camera was placed approx. 8 m from the side wall of the pool, at a height of approx. 5 m, in the mid-zone. This setting made it possible to record a distance of 10 m for each race length using a lens. As in the SAST, identification of this section was carried out on the basis of permanent markings on the edges of the pool.

### 2.4. Determination of Kinematic Variables

Both during stages 1 and 2, the Kinovea program (version 0.8.15, Joan Charmant & Contrib, Bordeaux, France) was used to determine kinematic variables. The time to cover each recorded 10-m section ($T_{10}$) was determined on the basis of the line joining the markings on the edges of the pool. The beginning and end of the 10-m swim was identified as the moment when the head crossed the designated lines (Figure 1). Based on the recordings, the duration of five cycles in the middle zone ($T_{5C}$) was also determined. The entry of the hand into the water was assumed as the beginning of the cycle. The same hand was dipped in the water again to complete the cycle. Based on the activities described above, for each covered section in the SAST and during the competition, the following were determined:

-    mean speed: $v = 10/T_{10}$ [m/s];
-    stroke rate: $SR = 60 \times 5/T_{5C}$ [cycle/min];
-    stroke length: $SL = v \times 5/T_{5C}$ [m];
-    stroke index: $SI = v \times SL$ [m$^2$/s].

The values of these variables from the SAST sample were used to later calculate the arithmetic mean ($\overline{x'}$), characterizing swimming during the initial ($v_S$, $SR_S$, $SL_S$, $SI_S$) as well as in the final parts ($v_F$, $SR_F$, $SL_F$, $SI_F$) of the test. The first group of indices was calculated on the basis of the results obtained in the 1st and 2nd measurement sections (10 m). The second type was calculated on the basis of the results obtained while swimming the 5th and 6th measurement sections of the SAST.

Additionally, the values of the fatigue indices were calculated in the SAST:

-    absolute: IFABS = SIS-SIF [dimensionless]

-    relative: IFREL = IFABS $\times$ 100/SIS [%]

Regarding the races, analogous variables were determined, except for the fatigue indices. In the 100-m freestyle swimming competition, the values of $v_S$, $SR_S$, $SL_S$, $SI_S$ were obtained on the basis of the recording the first 50-m section, while $v_F$, $SR_F$, $SL_F$, $SI_F$ were determined thanks to video recording analysis carried out swimming the second (final) 50-m section.

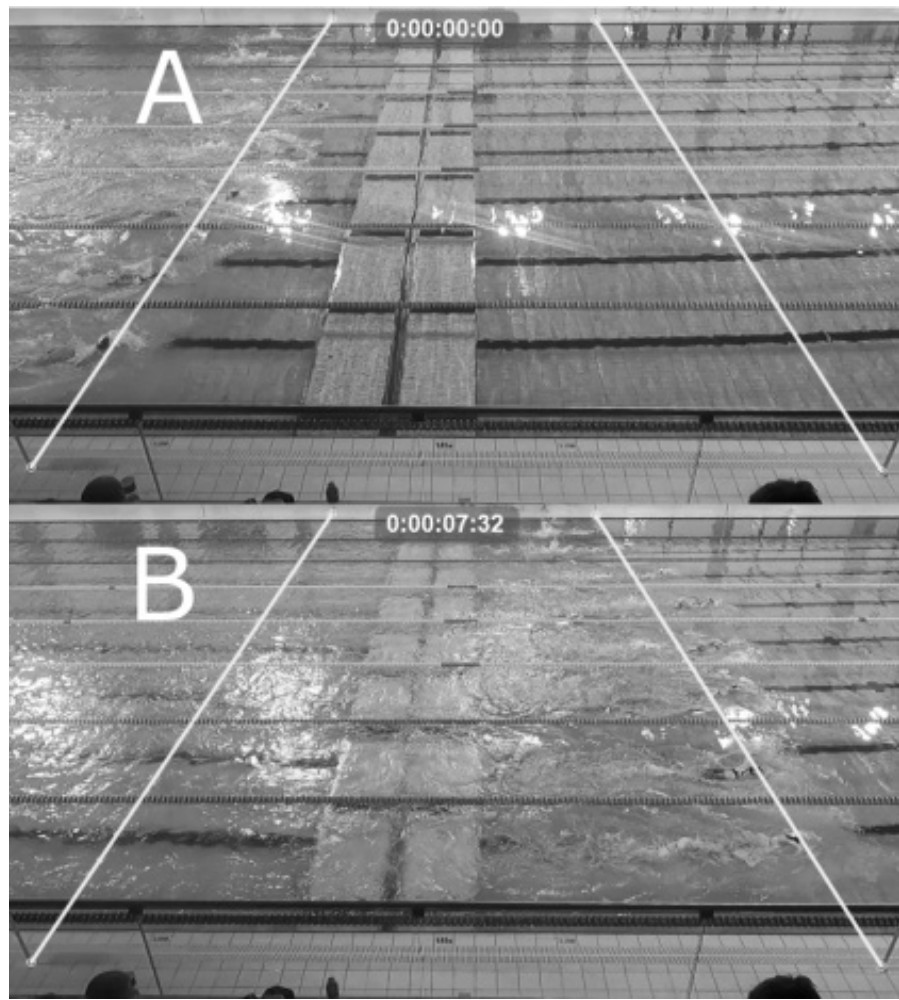

**Figure 1.** Method of determining time for 10 m covered by female competitor swimming the second lane from the bottom. (**A**) Start of covering distance, (**B**) completed distance.

In a similar manner, the unified variables in the 200-m freestyle swimming competition were also determined. The values of $v_S$, $SR_S$, $SL_S$, $SI_S$ were determined on the basis of the record of covering the first 50 m of the distance, while to determine $v_F$, $SR_F$, $SL_F$, $SI_F$, the record of covering the 4th length of the pool was used, i.e., the section between 150 m and 200 m of the distance. An additional source of data describing the performance of the tested swimmers at competitions were the official results available on the website: https://swimrankings.net (accessed on 6 October 2021). Further analyses included:

-    time for the 100-m race: T100 [s]
-    time for the 200-m race: T200 [s]

### 2.5. Statistical Calculations

The measurement data was analyzed using the Statistica program (version 13). For each variable, basic descriptive statistics were calculated. The next step was to perform one-way ANOVA (for samples of unequal size) to determine the significance of differences

in the values for the variables obtained in the SAST and for the 100- and 200-m freestyle races. The confidence interval limits for the means (95%) and partial eta-squared effect size ($\eta_p^2$) were determined. The interpretation of the effect size was conducted in accordance with that proposed by Lenhard and Lenhard [18]:

$0.01 \leq \eta_p^2 < 0.04$—the observed effect is small

$0.04 \leq \eta_p^2 < 0.14$—the observed effect is moderate

$0.14 \leq \eta_p^2 < 1.00$—the observed effect is large

In the next part of the statistical analysis, Tukey's RIR test was carried out for unequal numbers (post-hoc), which enabled the identification of variables for which the values significantly differed.

Next, a correlation matrix was created taking the SAST indicators into account as well as variables describing the performance of the participants at competitions. In the interpretation of the correlation coefficient (Pearson's *r*), assumptions were in accordance with Guilford and Fuchter's [19] proposal:

$0.5 \leq |r_s| < 0.7$—the observed correlation is high

$0.7 \leq |r_s| < 0.9$—the observed correlation is very high

$0.9 \leq |r_s| < 1$—the observed correlation is almost absolute

The significance of the correlation was found only for the minimal significance threshold of $p < 0.05$.

## 3. Results

The mean time ($\pm$SD) achieved during the 100- and 200-m freestyle races was: $T_{100} = 58.62 \pm 3.15$ s and $T_{200} = 130.33 \pm 7.88$ s, respectively.

In Table 1, means and limits of the confidence intervals for the means are presented for the kinematic variables from the initial sections of the SAST as well as the 100- and 200-m freestyle races, the variables from the final sections recorded in these trials, and the results of ANOVA aimed at seeking differences in the level of these indices.

**Table 1.** Descriptive characteristics ($\bar{x} \pm SD$ and the confidence limits for the means) of kinematic indices from the SAST test as well as the 100- and 200-m freestyle races together with the results of analysis of variance.

| Parameter | SAST | | 100 m | | 200 m | |
|---|---|---|---|---|---|---|
| | $\bar{x} \pm SD$ | Limits for the Mean | $\bar{x} \pm SD$ | Limits for the Mean | $\bar{x} \pm SD$ | Limits for the Mean |
| $v_S$ ♦,† [m/s] | $1.71 \pm 0.10$ | 1.66–1.77 | $1.70 \pm 0.09$ | 1.65–1.75 | $1.60 \pm 0.10$ | 1.55–1.65 |
| $v_F$ ♦,† [m/s] | $1.58 \pm 0.09$ | 1.53–1.62 | $1.57 \pm 0.09$ | 1.52–1.62 | $1.44 \pm 0.10$ | 1.39–1.49 |
| $SR_S$ ♦,† [cycle/min] | $53.46 \pm 3.46$ | 51.56–55.37 | $50.88 \pm 3.02$ | 49.22–52.54 | $42.72 \pm 3.21$ | 41.08–44.37 |
| $SR_F$ ♦,† [cycle/min] | $48.55 \pm 2.62$ | 47.10–49.99 | $46.84 \pm 3.45$ | 44.94–48.74 | $41.87 \pm 2.81$ | 40.44–43.31 |
| $SL_S$ ♦,† [m] | $1.93 \pm 0.13$ | 1.86–2.00 | $2.01 \pm 0.17$ | 1.92–2.11 | $2.26 \pm 0.19$ | 2.16–2.36 |
| $SL_F$ [m] | $1.95 \pm 0.13$ | 1.88–2.03 | $2.02 \pm 0.17$ | 1.93–2.11 | $2.07 \pm 0.13$ | 2.00–2.14 |
| $SI_S$ [m²/s] | $3.30 \pm 0.36$ | 3.11–3.50 | $3.44 \pm 0.44$ | 3.20–3.68 | $3.62 \pm 0.45$ | 3.39–3.85 |
| $SI_F$ [m²/s] | $3.09 \pm 0.35$ | 2.90–3.28 | $3.18 \pm 0.38$ | 2.97–3.38 | $2.99 \pm 0.34$ | 2.82–3.16 |

♦ indicates significant differences between SAST and 200-m race, $p < 0.02$. † indicates significant differences between 100- and 200-m races, $p < 0.02$.

As it turned out, the selected variables recorded during the SAST test and during the competition at shorter (100-m) and longer distances (200-m) differed significantly. The variables $v_S$ and $v_F$ were higher ($p < 0.001$) in the SAST and 100-m event than for the 200-m distance with a large effect size ($\eta_p^2 = 0.22$ and $0.32$, respectively). Higher values between the aforementioned tasks were also reported for $SR_S$ and $SR_F$ ($p < 0.001$) ($\eta_p^2 = 0.67$ and $0.48$, respectively). $SL_S$ was the only variable with lower values in the SAST and 100-m freestyle compared to the 200-m event ($\eta_p^2 = 0.42$, a large effect size).

Tukey's (post-hoc) test results are also shown in Table 1. It was observed that the mean values of most kinematic variables ($v_S$, $v_F$, $SR_S$, $SR_F$, $SL_S$) in the SAST as well as 100-m test are similar and, at the same time, significantly different from those recorded in the 200-m test. The $v_S$, $v_F$ and $SR_S$, $SR_F$ values recorded during the competition for the 200-m distance were significantly lower ($p < 0.05$) than the values noted in comparable measurements (SAST, 100 m). The initial stroke length ($SL_S$) in longer-distance swimming (200 m) was significantly higher ($p < 0.05$) compared to the remaining measurements.

The next step in data analysis was a search for correlations between the SAST results and the variables recorded during freestyle swimming competitions at 100 (Table 2) and 200 m (Table 3).

**Table 2.** Pearson's r correlation matrix between the SAST variables and 100-m freestyle.

| 100 m/SAST | $T_{100}$ | $v_S$ | $SR_S$ | $SL_S$ | $SI_S$ | $v_F$ | $SR_F$ | $SL_F$ | $SI_F$ |
|---|---|---|---|---|---|---|---|---|---|
| $v_S$ | −0.97 * | 0.91 * | ns. | 0.71 * | 0.85 * | 0.86 * | ns. | 0.66 * | 0.87 * |
| $SR_S$ | ns. | ns. | 0.55 * | ns. | ns. | ns. | 0.65 * | ns. | ns. |
| $SL_S$ | −0.51 * | 0.56 * | −0.64 * | 0.81 * | 0.76 * | ns. | −0.65 * | 0.88 * | 0.82 * |
| $SI_S$ | −0.83 * | 0.83 * | ns. | 0.88 * | 0.92 * | 0.72 * | ns. | 0.90 * | 0.97 * |
| $v_F$ | −0.96 * | 0.83 * | ns. | 0.63 * | 0.75 * | 0.92 * | ns. | 0.55 * | 0.83 |
| $SR_F$ | ns. | ns. | ns. | ns. | ns. | ns. | 0.54 * | ns. | ns. |
| $SL_F$ | −0.67 * | 0.66 * | ns. | 0.76 * | 0.77 * | 0.63 * | ns. | 0.76 * | 0.84 * |
| $SI_F$ | −0.87 * | 0.80 * | ns. | 0.78 * | 0.84 * | 0.83 * | ns. | 0.73 * | 0.92 |

* indicates $p < 0.05$; ns. indicates 'not significant' for $p < 0.05$.

**Table 3.** Pearson's r correlation matrix between the SAST variables and 200-m freestyle.

| 200 m/SAST | $T_{200}$ | $v_S$ | $SR_S$ | $SL_S$ | $SI_S$ | $v_F$ | $SR_F$ | $SL_F$ | $SI_F$ |
|---|---|---|---|---|---|---|---|---|---|
| $v_S$ | −0.95 * | 0.92 * | ns. | 0.65 * | 0.88 * | 0.83 * | ns. | 0.52 * | 0.79 * |
| $SR_S$ | −0.61 | 0.57 | ns. | ns. | ns. | 0.54 * | 0.67 * | ns. | ns. |
| $SL_S$ | ns. | ns. | −0.51 * | 0.65 * | 0.56 * | ns. | ns. | 0.72 * | 0.54 * |
| $SI_S$ | −0.75 | 0.74 | ns. | 0.80 * | 0.89 * | 0.67 * | ns. | 0.76 * | 0.83 * |
| $v_F$ | −0.85 * | 0.85 * | ns. | 0.50 * | 0.74 * | 0.69 * | ns. | 0.50 * | 0.68 * |
| $SR_F$ | ns. | ns. | 0.75 * | −0.61 | ns. | ns. | 0.78 * | −0.73 * | ns. |
| $SL_F$ | −0.48 * | 0.51 * | −0.50 * | 0.79 * | 0.78 * | ns. | ns. | 0.90 * | 0.74 * |
| $SI_F$ | −0.69 * | 0.71 * | ns. | 0.74 * | 0.84 * | 0.55 * | ns. | 0.81 * | 0.78 * |

* indicates $p < 0.05$; ns. indicates 'not significant' for $p < 0.05$.

The time to cover the 100-m ($T_{100}$) distance was negatively correlated with all variables recorded during the SAST. Apart from $SR_S$ and $SR_F$, these correlations were high, very high, and almost absolute ($r$ from −0.51 to −0.97). The existence of significant ($p < 0.05$), positive correlations ($r > 0$) between all unambiguous variables, the values of which were recorded in the SAST and during the 100-m swim, are also shown. The strongest correlations ($r$ from 0.81 to 0.91) were speed ($v_S$ and $v_F$), stroke index ($SI_S$ and $SI_F$), and stroke length ($SL_S$ and $SL_F$). The correlations between the frequency of movements ($SR_S$ and $SR_F$) showed a weaker correlation ($r$ approx. 0.55).

The values of Pearson's r correlation coefficients between the variables recorded in the SAST and their counterparts obtained during the 200-m freestyle competition are presented in Table 3. The majority of the SAST variables formed strong, negative, and significant ($p < 0.05$) correlations with the 200-m swim time ($T_{200}$). Besides one exception ($SR_S$), the correlations between the remaining unambiguous variables recorded in the SAST and the 200-m swim were high and very high ($r$ between 0.65 and 0.92, $p < 0.05$).

As described in the part of the work devoted to methodology, the values of the SI variable from the initial and final parts of the SAST were used to determine absolute fatigue index ($IF_{ABS}$) and relative fatigue index ($IF_{REL}$). The mean values of these variables were: $0.22 \pm 0.13$ m$^2$/s and $6.52 \pm 3.58\%$, respectively. It was found that none of these variables were related to the kinematic variables characterizing the 100- and 200-m freestyle races.

## 4. Discussion

The main findings of our study were the close relationship between the SAST results and the 100-m freestyle race. The values of indices from the initial and final parts of these motor tasks demonstrated similar values, and the correlation coefficients between them were from high to almost absolute. However, a smaller similarity was noted in relation to the kinematic indices for the 200-m freestyle race. At the same time, it has not been shown that the determined fatigue indices were related to the results for short and middle distances.

Among the determined indices from the SAST sample, six of them ($v_S$, $SL_S$, $SI_S$, $v_F$, $SL_F$, $SI_F$) were significantly negatively correlated with the result for 100 m. The only variables unrelated to time during the 100-m distance were $SR_S$ and $SR_F$. The variables from the SAST, determined on the basis of the times needed to cover the initial and final sections, i.e., $v_S$, $v_F$, $SI_S$, $SI_F$, demonstrated particularly high correlation coefficients. This shows high similarity in the movement patterns for the SAST and 100-m freestyle race. Slightly lower but significant correlations were also noted for the indices from this test and the 200-m freestyle race. In this case, $v_S$, $v_F$, and $SI_S$ had the highest coefficients. Smaller but still significant correlations were recorded in relation to $SI_F$, $SR_S$ as well as $SL_F$.

In the past, relationships were searched for between the results of tests based on covering a given distance several times and the results in various swimming races [14,20–22]. The literature in this field is primarily focused on series aimed at assessing aerobic capacity and their usefulness within the context of predicting the outcome for middle or long distances. Such tests, aimed at assessing anaerobic capacity, should be performed at maximum intensity, with the duration of the task short, while the interval between repetitions should not be long. Some of the tests of this type carried out so far [23] do not meet the above given criteria. To the best our knowledge, the evaluation of correlations between this type of test and the result for short distances was studied only by Meckel et al. [24]. However, they did not show that the 8 × 15 m test, with an interval of 30 s, was useful within the context of predicting the result for the 100-m freestyle. For this reason, the comparison of the numerical data obtained in this study to that available in the literature is limited.

In this study, a strong correlation was found between kinematic indices for the SAST and 100-m freestyle tests. The kinematic indices determined during the SAST showed similar values as during the 100-m freestyle swim. These concerned both the values of the first two ($v_S$, $SR_S$, $SL_S$, $SI_S$), as well as the final two repetitions for the 25-m sections ($v_F$, $SR_F$, $SL_F$, $SI_F$), as well as their equivalents from the first (0–50 m) and second (50–100 m) parts of the 100-m freestyle race. Small differences between the values of kinematic variables indicate a very high similarity in terms of swimming technique for the SAST and the short-distance race. This is also evidenced by the values of the correlation coefficients between all the variables of these two motor tasks. At the same time, some of the indices determined in the SAST and during the 100-m freestyle race ($v_S$, $v_F$, $SR_S$, $SR_F$, $SL_S$) differed significantly from those describing the 200-m freestyle. The probable cause for this was effort duration during the 100- ($T_{100}$ = 58.62 ± 3.15 s) and 200-m freestyle races ($T_{200}$ = 130.33 ± 7.88 s), which influenced the different contribution of aerobic and anaerobic changes in the supply into energy during these motor tasks. According to the data presented by Rodriguez and Mader [4], during a 100-m race, an athlete draws 59% of energy from anaerobic sources (including 20% phosphagenic) and 41% from oxygen, while in the 200-m race, these proportions are almost the opposite—42% anaerobic (including 13% phosphagenic) and 58% aerobic. Of course, it should be emphasized that the above-mentioned data relate to the top-level competitors for whom the duration of these races is shorter. In the case of younger swimmers, representing a lower sports level, the duration of the 100- and 200-m races is longer, thus, the aerobic component contribution may be greater [5]. This does not change the fact that, with regard to the 200-m competition, the SAST does not seem to be a sufficient tool for diagnosing the body's training status. For this reason, for swimmers specialising in middle-distance swimming, tests aimed at assessing aerobic capacity and power should be sought. It should also be mentioned that the differences in the values of

kinematic indices from the 100- and 200-m freestyle races may also be the result of different swimming strategies employed in these events. The swimmers start the 100-m race at almost maximum speed, which results in a 12% longer time to cover the second half of the distance [25]. This strategy is referred to as 'positive splitting' [26]. In the case of the 200-m distance, the intensity of the effort during the first 50 m is lower, which means that the increase in the time of completing subsequent sections is only 6–8% [25]. At this distance, most of the competitors also use the 'positive splitting' strategy, although some prefer 'even splitting' of pace, in which the time to complete following sections is similar [26].

In this study, no significant differences were found for stroke index measured during the initial and final sections of the SAST as well as the 100- and 200-m freestyle races ($SI_S$i $SI_F$, respectively). It was also noticed that other test variables such as speed, stroke length, or stroke rate differed much more in relation to these two distances. This means that they are modified to a greater extent depending on the duration of the effort (100 m and 200 m). At the same time, an exceptionally strong correlation was observed between the stroke index in the initial and final repetitions of the SAST and their 100-m equivalents ($r = 0.92$ for both $SI_S$ and $SI_S$) and the 200-m freestyle ($r = 0.89$ for $SI_S$ and $r = 0.78$ for $SI_S$). This indicates that the stroke index, in its assumptions reflecting technical efficiency [27,28], is characteristic of each athlete and is characterised by little variability depending on the conditions (e.g., distance length). This means that the stroke index indicators from the SAST can be used to assess the technical efficiency of athletes—both within the context of short and middle distances.

Improving sports results in many disciplines is possible by increasing the ability to continue exercise despite extreme neuromuscular fatigue [28,29]. Fatigue is associated with a decrease in the capacity to develop a force of an expected value [5], which in swimming, is mainly manifested by a reduction in speed during the final parts of a race [30,31]. Some authors also note a decrease in stroke length with increasing fatigue [30–32]. Conclusions from research concerning the influence of fatigue on stroke rate are not unequivocal. Bielec et al. [30], Toussaint et al. [33] and Barbosa et al. [34] showed that in such conditions, stroke rate decreases; however, in other studies [31,35], an increase in the value of this index has been noticed. The presented lack of clarity in the given sources may be the result of the selection of groups—in the research by Figueiredo et al. [30], the participants were world-class competitors, while Bielec et al. [30] made observations in a group of swimmers clearly representing a lower level (semi-competitive). In our study, both in the SAST test and the 100- as well as 200-m races, a decrease in speed was noted between the beginning and end parts of these motor tasks, while a reduction in the stroke length value was recorded only in the 200-m race. This means that the decrease in speed was more related with changes in stroke rate, because both in the SAST and 100-m race, a clear decrease in the value of this indicator was noted. It should be emphasized that the authors of the study involved young athletes who primarily experience central fatigue, while adults are dominated by its peripheral type [5]. This means that the generalization of the conclusions in this study to adult athlete groups is limited. At the same time, the comparison of data from the literature with our own results should also be carried out with caution.

In this study, based on the results of the SAST, two indices of fatigue (in absolute terms of $IF_{ABS}$ and in percentage of $IF_{REL}$) were determined, expecting that they would be correlated with the results for the 100-and 200-m freestyle distances. The results of research indicate that none of the indices were significantly related to the 100- and 200-m freestyle times. Similar results were obtained by Meckel et al. [24], finding no correlations between the size of performance decrement and the result for the 100-m freestyle. Apart from the earlier discussed possible impact of pace distribution on the speed of covering the 100- and 200-m distances [25], a probable cause could be the inter-individual differences in the aspect of skeletal muscle structure. Lievens et al. [36] indicated that athletes with a predominance of fast twitch fibres achieve high absolute values of mechanical power in tests aimed at assessing performance while, at the same time, recording large decrements. In turn, athletes with a predominance of slow twitch fibres in this type of tests are not able to achieve peak

values of mechanical power, but, at the same time, the evaluated performance decrement is small. This information is fully supported by the data provided by Mitchell et al. [22]. They found that short-distance (100 m) swimmers experienced greater performance decrements while performing a 12 × 25 m task than middle-distance (200 m) swimmers. Therefore, a further direction of research should be to track changes in the level of kinematic indicators in relation to various training cycles during the swimming season, taking the specific style of athletes into account.

**5. Conclusions**

The results of this study allow to indicate that there is a high similarity between the SAST and the 100-m freestyle result, and therefore, this test has potential in the context of predicting results for short distances. Simultaneously, the use of this test to predict the result at 200-m may be limited. The stroke index determined on the basis of the SAST sample may be a measure of technical effectiveness. At the same time, the fatigue index derived from the stroke index has limited use in comparing fatigue between individuals. The results of the research allow to indicate that this test can be used in coaching practice to monitor the fitness status of swimmers performing short- and middle-distances. The use of the Stroke Index in this test can be implemented to compare technical performance between competitors. Nevertheless, it is warranted to further evaluate the suitability of SAST, e.g., within the context of detecting individual training changes in different training cycles. It is also justified to investigate the relationship between the SAST and competition results with the use of regression analysis.

**Author Contributions:** Conceptualization, Ł.W. and R.S.; methodology, Ł.W. and R.S.; software, Ł.W.; validation, Ł.W; formal analysis, Ł.W., R.S. and M.S.; investigation, Ł.W., R.S. and M.S.; resources, Ł.W., R.S. and M.S.; data curation, R.S.; writing—original draft preparation, Ł.W., R.S. and M.S.; writing—review and editing, Ł.W., R.S. and M.S.; visualization, Ł.W.; supervision, Ł.W.; project administration, Ł.W.; funding acquisition, R.S. All authors have read and agreed to the published version of the manuscript.

**Funding:** Akademia Wychowania Fizycznego im. Bronisława Czecha w Krakowie (Award number(s): 022/RID/2018/19). The publication was financed within the program of the Minister of Science and Higher Education under the title "Regional Initiative of Excellence" (in Polish: "Regionalna Inicjatywa Doskonałości") within the years 2019–2022, project No. 022/RID/2018/19.

**Institutional Review Board Statement:** The study was conducted in accordance with the Declaration of Helsinki and approved by the Ethics Committee of Regional Medical Chamber (protocol code 172/KBL/OIL/2019 and date of 9 July 2019).

**Informed Consent Statement:** Informed consent was obtained from all subjects involved in the study.

**Data Availability Statement:** Not applicable.

**Conflicts of Interest:** The authors declare no conflict of interest.

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
