# Peer review of "Evaluating the Usefulness of the Modified Swimming Anaerobic Sprint Test (SAST) Based on the Relationship with the 100- and 200-m Freestyle"

_applsci, doi:10.3390/app12157566_

Round 1

Reviewer 1 Report

In this study the authors examine the relationship between the Modified Swimming Anaerobic Sprint Test (SAST) results and those obtained for the short (100-m style) and middle distances (200 m)..

Although the study has the potentiality of being shared with the scientific community, I believe that the manuscript would benefit from a major revision with the attempt to better support their experimental setting.

1.     Abstract: they should start with a first paragraph describing the background. This section should outline the following information:

-        What is already known about the subject, related to the paper in question

-        What is not known about the subject and hence what the study intended to examine (or what the paper seeks to present)

2.     The theoretical framework is scarce, they should clearly describe the scientific evidence that supports the hypothesis they have raised.

3.     Some necessary information is missing in methods section:

-        Experimental procedures should be better defined

-        The intervention protocol should be better described.

-        More information should be provided about the participants’ characteristics.

-        Anthropometric measurements and physical tests presuppose a protocol. This element is missing from the methodological description, which may imply an impossibility of replicating the study due to a lack of clarity in this regard.

-        What were inclusion and exclusion criteria?

4.     I would like to see more of the practical implications. Based on the analyzed variables, how the authors intend to use their findings?

Kind regards

Author Response

Dear Reviewer,

We are very grateful for the extremely substantive assessment of the article by the reviewer. Thanks to your efforts, the article gained significantly in value. We followed nearly all of the guidelines suggested by the reviewer. They include:

1. Abstract

- a short introduction has been added in order to familiarize with the current state of knowledge on the issue under study

2. Introduction:

- added information associated with the aim and hypotheses of the research: "Short efforts of maximum intensity meets the design criteria for the anaerobic capacity tests" (line numbers 65-66) and "Therefore, it seems reasonable to modify this test to the conditions of the aquatic environment and to determine the relationship between its results and the swimming result, especially at short and middle distances”(line numbers 70-72).

3. Materials and Methods:

- more detailed characteristics of the participants have been added

- this chapter was organized by adding the name of the first section "Participants and anthropometric measurements" and supplementing the description of anthropometric measurements

- in the statistical part, a description of the proposed procedure has been added - calculation of the effect size (partial eta-squared) and the limits of the confidence intervals of the means.

4. Results:

- the content was modified according to the statistical procedure - in table 1 includes the limits of the confidence intervals of the means, and the text adds effect sizes

- the description of the first part of the results was modified according to the comments (clarification of the terms "greater / less"

5. Conclusions:

- practical implications are included (line numbers 385-392)

As required, all corrections were made in the change tracking mode. We sincerely hope that the current version of the article is fully compliant with the publication requirements of Applied Sciences.

Yours faithfully, on behalf of the authors,

The main author

Reviewer 2 Report

Congratulations to the authors for preparing the study. The results found are interesting for the area of sport sciences, especially in relation to the assessment of performance in swimming. On the other hand, some issues, both general and specific, need improvement and/or explanation:

1)   the title and the aims use the term "predicting", but the statistical methods used do not allow to predict. Correlations do not predict, they only inform the degree of association between variances. In this case, regression analyzes would be more appropriate, or changing the title, objectives, discussion, and conclusion of the study.

2)   In the abstract (and along the text):

a.     age and body mass don`t need values with centesimal numbers (these are more associated with error than actual values). In this way, it should be 67.5 not 67.58 kg, e.g.

b.     Line 14: The variables obtained from the tests were compared, not the results.

c.      Line 17: The performance was identified as the time waste in the specific event, so use performance, not “time”,

d.     Line 18: ANOVA is analysis of variance, it`s redundant.

e.     Review the conclusion frase, please, regarding the “prediction” issue.

3)   Introduction:

a.     Line 33: “large body size”is not exactly the better option, since large body size increases drag, which limits swimming performance. Please rewrite.

b.     Line 43: What does mean “body`s efficiency”? Do you mean propulsive efficiency? Mechanical efficiency? Energy efficiency?

c.      Line 47: “Good indicator of stroke efficiency”...what does it mean? Arm stroke efficiency?

d.     Line 69: Do you mean 100 m freestyle, instead style?

e.     Regarding the SAST, do you consider to have used the previous test (4 x 50 m) proposed by Pelayo? (Pelayo, P., Mujika, I., Sidney, M., & Chatard, J.-C. 1996. Blood lactate recovery measurements, training, and performance during a 23-week period of competitive swimming. Eur J Appl Physiol Occup Physiol, 74(1), 107-113).  Tucher, G., Castro, F.S., Garrido, N., & Fernandes, R.J. (2018). Blood Lactate Estimation in Age Group after a Sprint Swimming Test. Journal of physical education and sport, 18, 1123.) Why would the SAST be better than the 4 x 50 m?

4)   Materials and methods

a.     Line 78: Was the sample size estimated?

b.     Lines 78 and 79: Age and body mass don`t need values with centesimal numbers.

c.      Line 95: Instead of “...swimming the crawl...”please use “performing the front crawl stroke”.

d.     Line 113: 100- or 200-m”. The swimmers did not participate in both, 100 and 200-m? How many in each one?

e.     Statistical calculations: In the descriptive, please calculate and present in the results the limits of the confidence intervals of the means. For all comparisons, calculate and present in the results the effects sizes (if multiple comparison, eta2, if pairwise comparisons, Edge`s gama ot Cohen`s d, with respective interpretations). If the aims are related to predict, multiple regression method should be used. In this case performance in 100-m and in 200-m would be the dependent variables, and the parameters from the SAST would be the predictors. Beta parameters, with multicolinearity results would be very usefull. Please verify this possibility.

5)   Results: Line 185 and so on: Write it more directly: instead of stating that there are differences, already establish them and put the categorized result of the effect sizes. For example: the variables v and SR were higher (p < 0.05) in the SAST than in the 100 and 200m events, with an average effect size.

6)   Discussion: Please, organize in same size paragraphs.  From line 248 to line 290 it is just one paragraph. If the statistical methods change to Regression Analyzes, the discussion will deeply change.

Author Response

Dear Reviewer,

We are very grateful for the extremely substantive assessment of the article by the reviewer. Thanks to your efforts, the article gained significantly in value. We followed nearly all of the guidelines suggested by the reviewer. They include:

  1. Title:

- in line with the review, the term 'predicting' has been removed from it. The same was done in almost all the text. In some places the term has remained, but each time "may" has been added. At the same time, in the chapter „Conclusions” we added that a further direction of the research should be to perform a regression analysis between the SAST test and the 100 m and 200 m races. We would like to point out that we gave up the regression analysis. We decided that the size of the group of participants was insufficient to perform such statistical proceeding.

  1. Abstract:

- centesimal numbers of age and body mass were removed (the same was done in the Materials and Methods)

- line numbers 16-17 - the word "results" has been dropped and the word "time" has been replaced with "performance"

- the description of the analysis of variance has been simplified (line number 22)

  1. Introduction:

- to define "large body size" (line number 39) appropriate interpretation has been added in brackets

- the term "body's efficency" referred to the conditioning of swimmers (line number 48). It was replaced by the words "the swimmer's physical fitness"

- the term "stroke efficency" has been changed to "swimming efficency" (line number 53) according to the literature, e.g. Barbosa T. et al. 2010. Energetics and biomechanics as determining factors of swimming performance: Updating the state of the art, Journal of Science and Medicine in Sport 13: "SI (Stroke Index) can also be used as overall swimming efficiency estimation"

- line number 78 - "style" replaced with "freestyle"

  1. Materials and Methods:

- line number 111 - changed from "swimming the crawl" to "performing the front crawl"

- line number 129 - replaced "or" with "and"

- in the statistical part, a description of the proposed procedure has been added - calculation of the effect size (partial eta-squared) and the limits of the confidence intervals of the means.

  1. Results:

- the content was modified according to the statistical procedure - in table 1 includes the limits of the confidence intervals of the means, and the text adds effect sizes

- the description of the first part of the results was modified according to the comments (clarification of the terms "greater / less")

  1. Discussion:

- paragraph 3 was divided into two smaller ones (286-289), slightly modifying the text.

7. References:

- one reference title was added (line numbers 454-455) and the numbering of the subsequent ones was modified accordingly (it was also included in the text).

In the review, the question was asked whether the sample size was estimated. We did not make any calculations in this regard. You also asked if we were considering using the 4 x 50 m test (Pelayo et al. 1996, Tucher et al. 2018). The aim of our research was to use a test that is as similar to the RAST test as possible. Therefore, the number of repetitions was 6, the interval between them was 10 seconds and the distances were kept as short as possible for use (25m pool). When choosing 50 m sections, it would take an average of 25-30 seconds to complete each of them, which would be significantly different from the duration time of the RAST test. We would like to point out that our intention was to evaluate the indicators describing the technique of full-stroke swimming. In the case choosing distances of a 50 m, a turn could have an impact on the SAST test result - e.g. by performing underwater dolphin movements or modifying the Stroke Length and Stroke Rate immediately after surfacing (this has been observed in literaturę, e.g. Escobar S. et al. 2018, Modeling stroking parameters in competitive sprint swimming: Understanding inter- and intra-lap variability to assess pacing management, Hum Mov Sci 61: 219-230). This does not change the fact that we consider the studies mentioned by the reviewer to be valuable, in particular from the point of view of the possibility of observing changes under the influence of such efforts.

As required, all corrections were made in the change tracking mode. We sincerely hope that the current version of the article is fully compliant with the publication requirements of Applied Sciences.

Yours faithfully, on behalf of the authors,

The main author

Round 2

Reviewer 2 Report

Congratulations on the study. All questions were adequately answered and changes that increased the quality of the study were duly carried out. In my view, the study is accepted for publication